# Exploring the Cloud Computing Implementation Drivers for Sustainable Construction Projects—A Structural Equation Modeling Approach

Ahmed Farouk Kineber [1,*]ᴵᴰ, Ayodeji Emmanuel Oke [2,*]ᴵᴰ, Ashraf Alyanbaawi [3],
Abdurrahman Salihu Abubakar [4]ᴵᴰ and Mohammed Magdy Hamed [5,6]ᴵᴰ

1    Department of Civil Engineering, College of Engineering in Al-Kharj, Prince Sattam Bin Abdulaziz University,
     Al-Kharj 11942, Saudi Arabia
2    Department of Quantity Surveying, Federal University of Technology, Akure 340106, Nigeria
3    College of Science and Computer Engineering, Taibah University, Yanbu 46411, Saudi Arabia
4    Department of Civil Engineering, Faculty of Engineering, Universiti Putra Malaysia, UPM, Serdang 43400,
     Selangor, Malaysia
5    Construction and Building Engineering Department, College of Engineering and Technology, Arab Academy
     for Science, Technology and Maritime Transport (AASTMT), B 2401 Smart Village, Giza 12577, Egypt
6    Department of Water and Environmental Engineering, School of Civil Engineering, Faculty of Engineering,
     Universiti Teknologi Malaysia (UTM), Skudia 81310, Johor, Malaysia
*    Correspondence: a.farouk.kineber@gmail.com or a.kineber@psau.edu.sa (A.F.K.);
     emayok@gmail.com (A.E.O.)

**Abstract:** Sustainability aspects should be adopted during all the decision-making stages of executing construction projects to gain maximum benefits without compromising the objective of such projects. Cloud computing has been a valuable tool for sustainable construction success in several countries over the last two decades. Cloud computing and its drivers have undoubtedly improved the sustainable success target of cost, quality, and time. However, cloud computing implementation in Nigeria's construction industry is minimal. Consequently, the study aims to generate a decision support model to support a cloud computing implementation by looking into the relationship between cloud computing drivers and construction activities in Nigeria. This study's data was obtained from previous literature and quantitatively augmented with a questionnaire survey. The data was obtained from questionnaires administered to one hundred and four construction practitioners in Lagos State. Thus, exploratory factor analysis (EFA) was used to validate the questionnaire survey results. However, to assess and validate the factors (drivers) constructed and analyze the relationships between cloud computing drivers and construction activities, partial least square structural equation modelling (PLS-SEM) method was used. An analysis of construction project activities was carried out through EFA, and it generated five main components: pre-contract stage, management, design and storage, estimation and communications, and finally, back-office activities. The study indicated that the implementation of cloud computing drivers had a significant impact on construction activities. The findings also revealed a weak relationship between cloud computing implementation and construction activities, with a 0.087 percent impact. Furthermore, the findings indicate that human satisfaction is the primary factor influencing cloud computing deployment, followed by organization, client acceptance, and industry-based factors. The significance of the findings can be used as a reference or standard for decision-makers to base their decisions on the cost efficiency of cloud computing and its capability to boost efficiency in the construction sector. This research contributes to current construction engineering management by enhancing knowledge of cloud computing implementation drivers and their implications on construction activities.

**Keywords:** cloud computing; sustainable success; construction projects; partial least square; structural equation modelling; Nigeria

## 1. Introduction

The construction industry is critical to growing emerging countries' economies [1]. It is always reinventing itself via government technology and innovative methods [2]. However, the construction sector has undergone significant changes to meet national economic demands [3]. As a result, many developing countries are turning to cloud computing (CC) to help them improve their financial processes [4]. However, in an ever-changing and urbanizing world, construction project allocation cannot adequately meet demand [5]. In addition, construction projects frequently face several time schedules delay [6–8]. In addition, the sector's widespread productivity problems may be traced back to its slow adoption of new technology [9–12]. Moreover, technological strategies must be developed to enforce the performance growth concept [13]. Furthermore, reliable data is required to execute all construction works to enhance the "success of construction activities" due to data fragmentation. Thus, the construction industry in many developing countries is not gaining adequate support from the government, society, and clients [14].

Whereas sustainable idea improvement is not new [15], it seems to play an ever more important role in many organizations [16,17]. Consequently, there is a critical need to improve the "overall sustainable success of construction projects" resource-efficient way [18]. Cloud computing is at pace with recent technological developments, and literature on this technology and its uses is widely available. Hence, it may be used successfully during the phases of planning and execution. Wolstenholme, et al. [19] argued that for construction practices to be effective, modern tools need to be used to automate the construction process, leading to success in construction practices. Therefore, enhancing the successful activities and tracking the construction project's progress is essential. Cloud computing will help achieve this proposed construction success by providing remote access to computing resources through the internet using information communication technology (ICT) tools and techniques worldwide

Cloud computing is meant to transform sustainable economic activities worldwide into a versatile approach for sustainable management cost via its inherent pay-per-use network, availability, scalability, and other characteristics [20]. The cost economization will be a performance measure for construction projects [21]. Computing applications are offered via shared server networks that simultaneously serve numerous employers [22]. Cloud computing technology can produce high-performance computing power by analyzing large amounts of IoT data and providing a valuable decision-making vision [23]. Small and medium-sized enterprises (SMEs) now have a new tool, cloud computing, to address a wide range of sustainability concerns, such as those related to finance [24]. It utilizes services as a product, paying only for what is required [25].

Although previous studies have addressed the advantages of cloud computing, a diminutive effort was made to assess the cloud implementation drivers in developing countries; thus, its adoption is still shallow [26,27]. Many organizations still do not understand how cloud computing can impact or improve their work [28]. Fang, et al. [29] argued that there are just a few cloud computing applications in the construction sector. In addition, there has been little study on particular IT services, such as cloud computing, which offer benefits that include enhancing relational skills across organizations and improving economic efficiency [30]. However, Zainon, et al. [31] claimed that the developments in cloud computing and their effect on the building industry must be implemented to overcome construction problems, and cloud computing can assist in this case. Consequently, it is necessary to investigate the key variables impacting cloud computing adoption [32].

This exploratory research outlined the main research question built on the obtained results. The research question is, "What are the relationships and drivers needed to implement cloud computing in the Nigerian construction industry?" Therefore, these relationships need to be examined, and cloud computing drivers too need to be identified [33]. Rockart [34] identifies the drivers as "areas where, if satisfactory, the results will ensure the organization's competitive success." Likewise, Chan, et al. [35] and Yu, et al. [36] agree that the drivers should be seen as important management readiness and action in different

construction domains to bring about improvements [37]. By being mindful of these drivers, a firm may favorably impact the success of the development process while successfully mitigating its risks [38].

This research set out to learn which factors in Nigeria's building sector were responsible for its rapid uptake. The current research presents a novel attempt to fill this gap using the partial least square (PLS) modeling method to mathematically analyze the relationship between the implementation of cloud computing drivers and construction activities. It is noteworthy that this study used the global-local context (GLC) approach, which emphasized the study's worldwide importance. In addition, it signifies and magnifies the problems examined. Summers [39] adds that establishing the significance of a study in both a local and a broader context is a good method to market its significance. Because of this need for precision, the research will focus on "emerging" nations and, more specifically, on Nigeria as its local context (i.e., establishing the importance). Consequently, this study would be beneficial by assisting decision-makers to attain a successful construction project by reducing unnecessary costs and improving efficiency through cloud computing implementation in Nigeria and other underdeveloped nations where similar construction initiatives are being undertaken [40]. Many stakeholders in the construction industry, including policymakers, contractors, and designers, stand to gain from this research [41].

## 2. Cloud Computing-Related Implementation Literature

The phrase "cloud computing" refers to internet-based technology that stores information in servers and provides software as a service (SaaS) to users on-demand [32]. It has a substantial impact on clients and companies, so clients can access their data and information from any device, while organizations can rent computing resources (including software and hardware) and space storage from cloud service providers [42]. Therefore, it is assumed to be a beneficial way for companies to save money on IT, use less allocated space, lessen electricity consumption, boost efficiency delivery, provide added value, aid in job creation, and reduce the risk related to maintaining and managing the hardware infrastructure [43,44]. According to the National Institute of Standards and Technology's (NIST) definition, "cloud computing is a model that enables ubiquitous, convenient, on-demand network access to a shared pool of configurable computing resources (e.g., networks, servers, storage, applications, and services) that can be rapidly provisioned and released with minimal management effort or service provider interaction [45]." Since its inception in 2006, it has emerged as one of the leading technologies being explored for deployment by companies worldwide [46].

NIST defined cloud computing more precisely as "a model for enabling ubiquitous, convenient, on-demand network access to a shared pool of configurable computing resources (e.g., networks, servers, storage, applications, and services) that can be rapidly provisioned and released with minimal management effort or service provider interaction [47]." Cloud computing (CC) providers deliver all IT services on demand, while the payment is made for computer services and equipment [48]. There are three distinct types of cloud services: "Infrastructure as a Service (IaaS), Platform as a Service (PaaS), and Software as a Service (SaaS)". The CC service providers offer clients on-demand with basic computational capabilities in IaaS. [32]. Unlike ordinary hosting services, IaaS provides the ability to meet the varying needs of several users. As a result, it offers significant flexibility and cost savings compared to conventional computing technologies [49]. Service providers advertise their software programs through the internet under SaaS, while in other traditional IT solutions, installation of software programs is required. However, internet service companies charge for their services [49]. PaaS service providers present developers with solutions that are pretty superior to conventional workstation settings. It enables independent software providers and IT professionals to quickly create and deploy web applications using third-party infrastructure [32]. PaaS is a comprehensive platform for creating, designing, testing, and deploying a product. The PaaS users may build applications utilizing provisioner-supported APIs and programming languages and deploy the apps

immediately onto the cloud provider servers [50]. Examples include "Google (mail and drive) Drop and Zoho box, Office Live, iCloud, Yahoo, IBM and Adobe Creative Cloud as a platform and tools Service" [24,50–54].

Cloud computing services are used in many different "applications," such as remote forensics, hospitality, eGovernment, human resource management, and the Internet of Cars, as well as the processing of genomic data, teaching and learning, the services for small and medium-sized businesses, eLearning methods, manufacturing, emergency recovery, smart cities, and many others [55]. Recently, cloud computing has seen a meteoric rise in popularity globally [56]. When these qualities are used to their greatest extent, they can improve construction activities differently and are critical to one another [57]. As a result, academics are now investigating how cloud services collaboration may be used to create and synthesize data that will generate additional value to the potential strength of cloud computing systems and services. Figure 1, derived from [58], depicts the dynamics of the computer system, the application layer, the network layer, and the sensor layer.

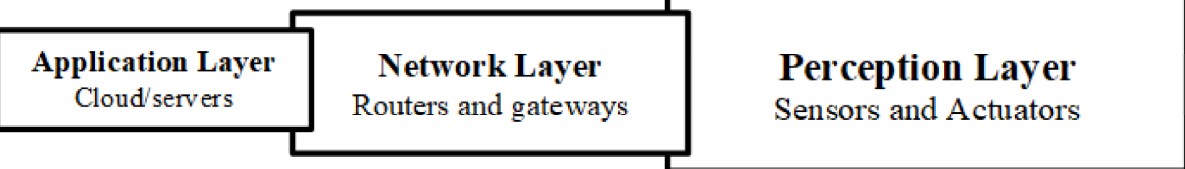

**Figure 1.** Cloud architecture dynamics.

Objects data can be read and stored from various platforms using visualization protocols. Thus, the processing burden can be reduced, and the information may also be analyzed on the cloud [59]. Using this analogy, the application layer might detect things within the surrounding environment while submitting queries to the cloud to analyze and sensor the data. Object information is reposted using data obtained from the sensor layer, and therefore data analysis is scheduled for later activities [60]. Subsequently, cloud computing enables commercial organizations to focus on their core processes, markets, and product innovation. A company's information technology department can now invest in other productive projects with all of the money, time, and effort that would have been spent on the IT department. Consequently, it enables businesses to use their precious and limited resources better to enhance their goods and services [24].

Cloud computing is divided into private, public, and hybrid clouds [61]. Private clouds are established within the company firewall; thus, the clouds are internal and may be accessed by the company's various divisions or departments [62]. In the public situation, clouds are designed and built outside the company's firewall [61]. Finally, hybrid clouds combine both public and private cloud features. Many companies install hybrid clouds to reap the advantages of the public cloud while still benefiting from the information security of private clouds [61].

The technology's significant features include "on-demand self-service, extensive network access, dynamic resource, quick flexibility or advancement, and measurable service" [45]. Since when the phrase "cloud computing" was first introduced in 2006 [63], it has received a massive investment of 266.4 billion USD with an average growth rate of 17% per annum and with approximately 60% of all companies projected to use an outsourced cloud service provider to install the system [63]. In the United States and Europe, the impact of cloud adoption has been well-researched as a vital infrastructure that enables governments to store, exchange, and analyze data to enhance existing services or develop new technologies and provide new services [64,65]. Hence, it is obvious that cloud computing adoption has several advantages. However, cloud companies generally face obstacles. For instance, in Europe, "culture, environment, legislation, economy and politics, IT personnel scarcity and sense of uncertainty, anxiety and lack of patience" were some of the significant obstacles observed [66]. Thus, this study is motivated by the need to look at developing

nations like Nigeria to better understand the full impact of cloud computing. The study was built on the belief that the environment plays a crucial influence in cloud acceptance, with countries in the developed regions being at a higher level than those in the developing regions [67]. While the United States and Europe have experienced higher levels of cloud acceptance, Asian and many other African countries are only beginning to accept it [67]. For example, Japan, South Korea, and Singapore are currently transferring public services into the cloud [63] and investing in developing national cloud infrastructure [68]. The rising cloud acceptance in the public and private sectors indicates its wide acceptance in these nations. For instance, Japan has a sophisticated cloud infrastructure, South Korea embraces cloud services as an essential component of the Industry 4.0 plan of the country, while Singapore provides accessible cloud services, and in Nigeria, it is yet to be fully explored [63]. Decision-makers find it more challenging to satisfy public expectations regarding quality, timeliness, innovation, and easy access to public services [69,70].

Cloud adoption overcomes this problem by reducing the time accessing public services, reducing logistic costs, and quality improvement in services through cloud computing services [71]. It also enables public services to be accessed at any time and location via mobile devices. According to the literature, cloud computing users see it as a breakthrough in adopting new technology and innovation [72]. It may also lead to accepting other innovative technologies, such as digital transformation in other sectors [73] and an integrative technology solution [74]. In addition, other developing countries, especially in Asia, are quickly adopting cloud technologies [75], given the region's growing demand for services to its citizens [76].

Various organizations are increasingly shifting to cloud computing since it delivers dynamic and scalable resources through online services [77]. As a result, cloud computing usage in the construction sector has seen a significant breakthrough over the last three decades, as shown by Table 1. However, research on cloud computing's actual use and implementation by construction players is wanting, particularly in developing nations like Nigeria [25]. Therefore, this study attempts to fill this gap by evaluating the relationship between the implementation drivers of cloud computing and construction activities to achieve optimal project delivery.

**Table 1.** Cloud computing drivers.

| Factor Component | Code | Drivers | Studies |
|---|---|---|---|
| Human satisfaction | D.1 | Availability | [50,51] |
| | D.2 | Privacy | [20,78] |
| | D.3 | Perceived ease of use | [51,78,79] |
| | D.4 | Performance | [51,54] |
| | D.5 | Cost of accessibility | [20,54] |
| | D.6 | Reliable data storage | [24,50,54] |
| Organization factor | D.7 | Size of organization | [20,24,54,79] |
| | D.8 | Capacity of organization | [20,24,54,79] |
| | D.9 | Structural template | [20,24,54,79] |
| | D.10 | Specialized human resources | [20,24,54,79] |
| Clients acceptance | D.11 | Willingness of clients | [50,51] |
| | D.12 | Clients' readiness | [51,78] |
| Industry-based factor | D.13 | Competitive pressure from within the industry | [20,24,79] |
| | D.14 | Nature of industry | [20,24,51,78–80] |
| | D.15 | Technological advancement | [24,50,51,54,79,80] |

## 3. Research Model Development

Recent expansion in the building sector has prompted research toward effective implementations of cloud computing in conventional workplaces [81]. The broad adoption of a strong CSR ethic across businesses may also boost the usage of cloud computing in important key sectors. In contrast to conventional methodologies and procedures,

Muhammad Abedi [82] argued that introducing cloud computing into construction projects would ensure collaboration with other construction stakeholders, enhance communication, and increase the feasibility of working together professionally. Other building partners can store and retrieve development data in real-time by working collaboratively through the cloud platforms. Studies have shown that mobile devices like laptops and personal digital assistants can improve on-site real-time data collection [25]. Mobile applications developed to monitor the project's sustainable development programs have proven this. Hence, a construction plan may be imported into a Microsoft Excel spreadsheet through software, and all other construction professionals in the office or at any other venue may have access to it and operate on any given task on the construction site. The on-demand and self-services features are part of the beneficial aspects of cloud computing services. Most computing paradigms are less flexible than cloud computing; unlike traditional storage models, cloud computing is a stand-alone entity, allowing users to link and use the technology from wherever they are via internet access [24].

As an added bonus, cloud computing frees up smaller enterprises to focus on what matters: making money and developing ground-breaking new products and services. Businesses must show that they have created firm value by creating a successful business strategy [83]. There is no longer any need to dedicate time and effort to the IT division; instead, those assets may be redistributed to more pressing business needs. Firms of all sizes whose core business is not IT infrastructure development can relax about the necessity of regular system maintenance and upgrades (IS). Instead, companies need to focus on what they do best in order to grow as a company and remain competitive. In addition, it motivates them to be creative and open to trying new things to find niches in the market [24]. Afolabi [53] claims that cloud computing aids businesses in reaching their productivity goals by reducing costs, improving efficiency, and enhancing the quality and efficiency of operations while allowing them to focus on what they do best rather than spending time and resources on tasks like technology management and upgrades.

The cost structure of cloud computing is an intriguing aspect of the technology. One related consideration is that cloud computing lowers customers' investment costs because users only pay for the resources they utilize [54]. There is little doubt that the pay-as-you-go model and scalability capabilities of cloud computing provide genuine benefits to cloud customers by reducing the upfront cost of putting up IT infrastructure services and the ongoing cost of maintaining them. These are substantial concerns that must be addressed before small and medium businesses in sub-Saharan Africa adopt cloud computing safely and successfully. Cloud computing reduces operating costs and personnel remunerations. It has also lowered the licensing rates for SMEs when purchasing software from tech vendors while concurrently reducing the cost of installing extensive computing facilities that rely heavily on energy [26]. Energy in developing countries is not usually available due to the precarious nature of power in many parts of Africa. However, it can be partially mitigated by using other energy sources like solar [84]. Cloud computing is an independent system that allows concurrent technology to run on various computers regardless of the local hardware used for the applications, making it essential to cloud computing services [26]. Cloud computing drivers are practices and procedures that must be followed to ensure that construction industry stakeholders effectively apply and implement quality management [85]. This research hypothesized a significant relationship between cloud computing driver implementation and construction activities based on the abovementioned literature, as shown in Figure 2. The first scale is shown in Table 1 based on drivers from previous studies [25]. Table 2 displays the construction activities that have the most significant impact on the construction industry.

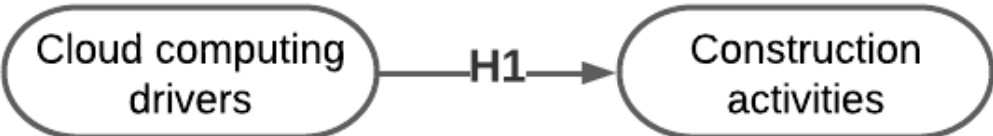

**Figure 2.** Influence of cloud computing drivers on construction activities.

**Table 2.** Construction activities.

| Activities | Code | [53] | [79] | [20] | [86] | [87] | [24] | [78] | [88] | [80] | [89] | [90] | [50] | [91] |
|---|---|---|---|---|---|---|---|---|---|---|---|---|---|---|
| Documentation | C1 | ✓ | ✓ | ✓ | ✓ | ✓ | ✓ | ✓ | ✓ | ✓ | ✓ | ✓ |  | ✓ |
| Project planning monitoring and modification | C2 | ✓ |  |  | ✓ | ✓ |  | ✓ |  | ✓ |  | ✓ |  |  |
| Architectural design | C3 |  |  |  |  | ✓ |  |  |  |  |  |  |  |  |
| Structural analysis | C4 | ✓ |  |  |  |  |  |  |  |  |  |  |  |  |
| Data storage | C5 |  |  | ✓ |  |  |  |  | ✓ |  |  |  |  |  |
| Procurement management | C6 | ✓ |  |  |  |  |  |  |  |  |  |  |  |  |
| Construction Management | C7 | ✓ |  |  |  |  | ✓ |  |  |  | ✓ |  |  |  |
| On-site information retrieval | C8 | ✓ |  | ✓ | ✓ | ✓ |  | ✓ | ✓ | ✓ |  | ✓ |  | ✓ |
| Cost estimation | C9 | ✓ |  |  |  |  |  |  |  |  |  |  |  |  |
| Communication | C10 |  |  |  |  |  |  |  |  |  |  |  | ✓ | ✓ |
| Logistics | C11 |  |  | ✓ |  |  |  |  |  |  | ✓ |  |  | ✓ |
| Back-office activities | C912 |  |  |  | ✓ |  | ✓ |  |  |  |  |  |  |  |

### 3.1. Subsection

A conceptual model is the first step in developing a research strategy. A conceptual model describes the topic graphically based on the literature review to produce intermediate theories (hypotheses) that can be tested based on empirical evidence [92]. The conceptual modelling phase is divided into three stages: (1) defining the model's constructs, (2) categorizing the constructs, and (3) determining the relationships between them [93]. The process elucidates the model's results, as shown in Figure 2. Furthermore, the research design was adopted from Kineber, et al. [94], as indicated in Figure 3.

### 3.1.1. Construct Validity Analysis

As previously stated, the constructs classifications for cloud computing drivers are based on the categorization of [25]. Furthermore, to classify the constructs related to construction activities (Table 2), the previous literature was critically reviewed to identify the critical construction activities and drivers, using exploratory factor analysis (EFA) to analyze the groups. EFA was also used to evaluate the constructs' validity by estimating the non-dimensionality, reliability, and validity of each construct's measurement components (i.e., the measurement models). "It is worth noting that Principal Component Analysis (PCA) was chosen over other approaches since PCA as a more reliable and less conceptually complex method" [95]. In addition, varimax rotation was used rather than direct noblemen or promax because varimax rotation increases load dispersion among variables [96]. Consequently, factor analysis was carried out on the 15 defined factors based on the questionnaires obtained from 104 participants in the current study [97].

### 3.1.2. Analytical Approach (Structured Equation Modelling)

The SEM analysis was performed to evaluate the influence of cloud computing drivers on construction activities [98,99]. The SEM method was chosen because it gives the relationship between many observable and non-observable variables; hence SEM was appropriate for this analysis [100,101]. The PLS-SEM method was used in this analysis to create the model and determine the relationship between cloud computing drivers and construction activities. PLS-SEM has become a proven non-experimental research technique where hypothesis testing methods are complicated [102,103]. Yuan, et al. [104] said that PLS-SEM is a well-known analysis method and the most widely adopted form of data analysis in

social sciences. It also has a wide-ranging statistical scope that could be used to assess the measurement and structural models [105,106]. Therefore, this method was used in this research as it can be used to analyze the data obtained from the construction industry [18,107]. Moreover, it is a forecast-oriented evaluation tool that can deal with complex data [108,109].

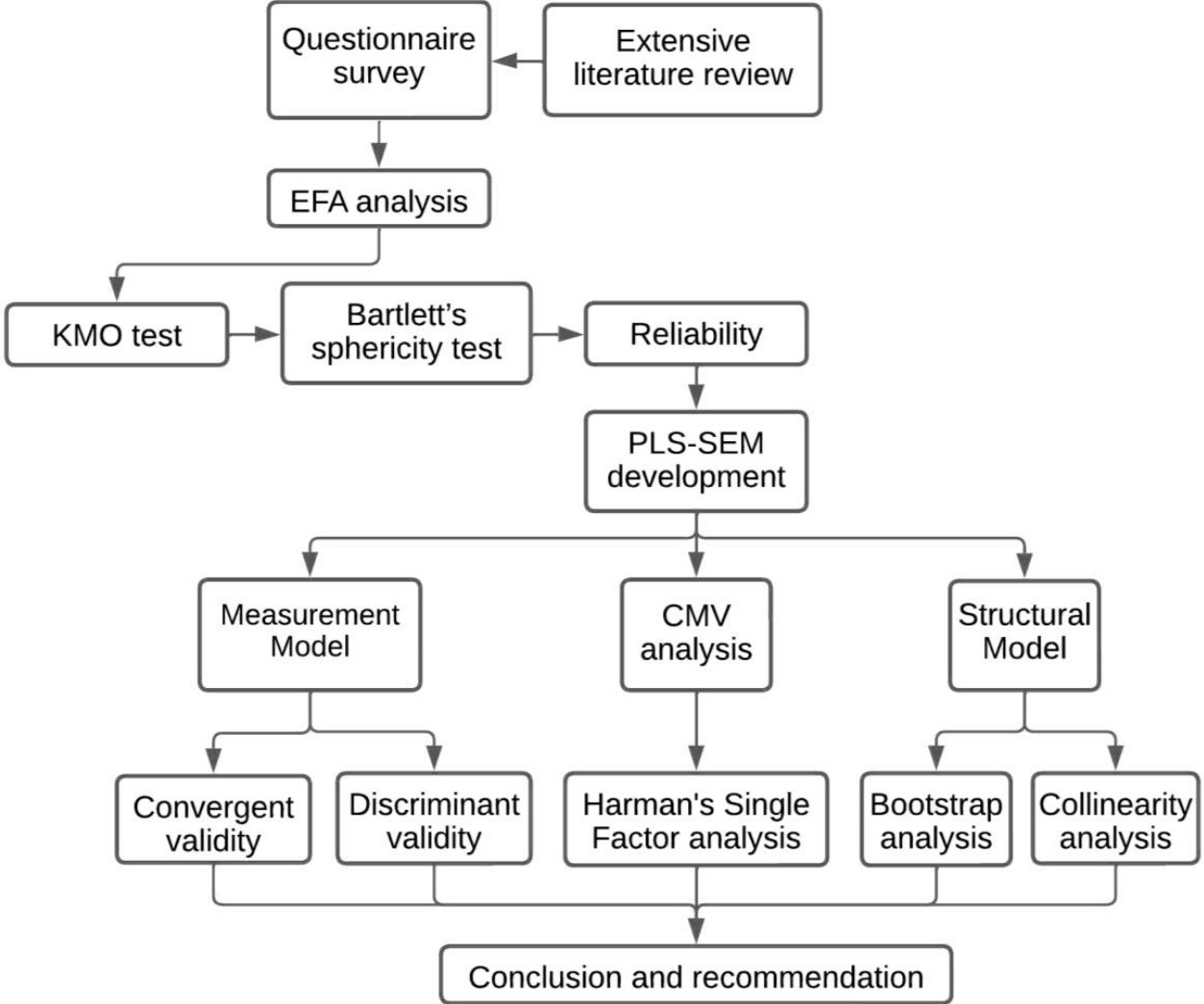

**Figure 3.** Research design.

### 3.1.3. Data Collection

This study uses cloud computing drivers to efficiently execute construction projects in the Nigerian construction industry. The data was collected from the target population using a simple random sampling technique. The sample size was determined based on the targeted population [110]. Consequently, a suitable statistical analysis technique was chosen to generate the proposed model based on the sample size. SEM was selected for this study; the sample size should be sufficient to achieve the desired result and offer an alternative model [18]. For SEM, Yin [111] agreed that the sample size should be greater than 100.

However, many other researchers oppose maximization and recommend optimizing the sample size [112]. They argued that it was not cost-effective and time-efficient at a certain level; however, its generalizability is a significant sample size advantage. Therefore, the minimum measured sample size was taken to achieve the desired statistical power level [113,114]. The SEM requires a suitable sample size to acquire consistent estimations [115]. Gorsuch [116] suggested a minimum of five participants for every construct and one hundred individuals for every data analysis.

Accordingly, the PLS-SEM analysis of the study was the chosen option over the covariance-based SEM (CB-SEM) method because it best fit the analysis structure of the study. PLS-SEM can be used to evade constricting assumptions that form a total estimate of the total possible deductions with a minimum sample size [117–120]. The sample size for conducting PLS requires only 30–100 responses, as mentioned by [119,121]. Consequently, 137 questionnaires were distributed, and 104 responses were returned and accepted, representing 76 percent of the total population and falling within the acceptable range for analysis [97]. The achieved sample size also meets the minimum numbers recommended, and this tallies the minimum number of sample sizes required [111,116,119–121]. Furthermore, the sample size used in this research is similar to that used in a study on applying PLS-SEM in building projects. It suggests that the data gathered was sufficient for future empirical testing [122].

The survey questionnaire was divided into four sections. The first section of the questionnaire collects the demographic data of the respondents. Furthermore, the second section was used to collect responses on cloud computing drivers (Table 1), and the third section was used to collect data on construction activities (Table 2) using " a five-point Likert scale with 5 = very high, 4 = high, 3 = average, 2 = low, and 1 = very low," as used in many previous studies [18,123–128]. Finally, there were open-ended questions to include additional activities or factors that the respondents believed should be included.

## 4. Results

### 4.1. Respondents Background

This section presents the respondents' demographic background, essential characteristics, and knowledge of cloud computing technologies. Figure 4 shows that the majority of the firms had between 6 and 10 years of operational experience in the industry, with 28 responses (26.9%), 11 to 15 years having 24 responses (23.1%), those with less than 5 years' experience having 10 responses (13.5%), 15 to 20 years with 9 responses (8.7%), and more than 21 years having 1 response (1.0%). Thus, most registered construction companies have 6 to 10 years of operational experience. Under the professional variables, quantity surveyors had the most significant percentage (39.4%) with 41 professionals, while architects accounted for (26%), and engineers for (21.2%) of the total data collected. In terms of registration with professional regulatory bodies, 27 of the respondents representing 26% were licensed members with "QSRBN (Quantity Surveyors Registration Board of Nigeria)", 12.5% were licensed with "CORBON (Council of Registered Builders of Nigeria)", 24% were certified by "ARCON (Architects Registration Council of Nigeria)", and 21% were certified engineers by "COREN (Council for the Regulation Engineers of Nigeria)." The results also reveal the respondents' educational qualification, indicating that a higher percentage of the respondents had a bachelor's degree (BTech/BSc) with 46 responses (44.2%), a master's degree (MSc/MTech) with 39 responses (37.5%), PhD holders had 13 responses (12.5%), and HND (higher national Diploma) holders had 6 responses (5.8%). The results showed that construction practitioners with cloud computing knowledge were considerable, as shown in Table 2. It indicates that 95.2% have been actively engaged in cloud computing operations. Cloud computing familiarity is relatively high, as shown in Figure 4. 96.2% of construction practitioners were aware of cloud computing initiatives.

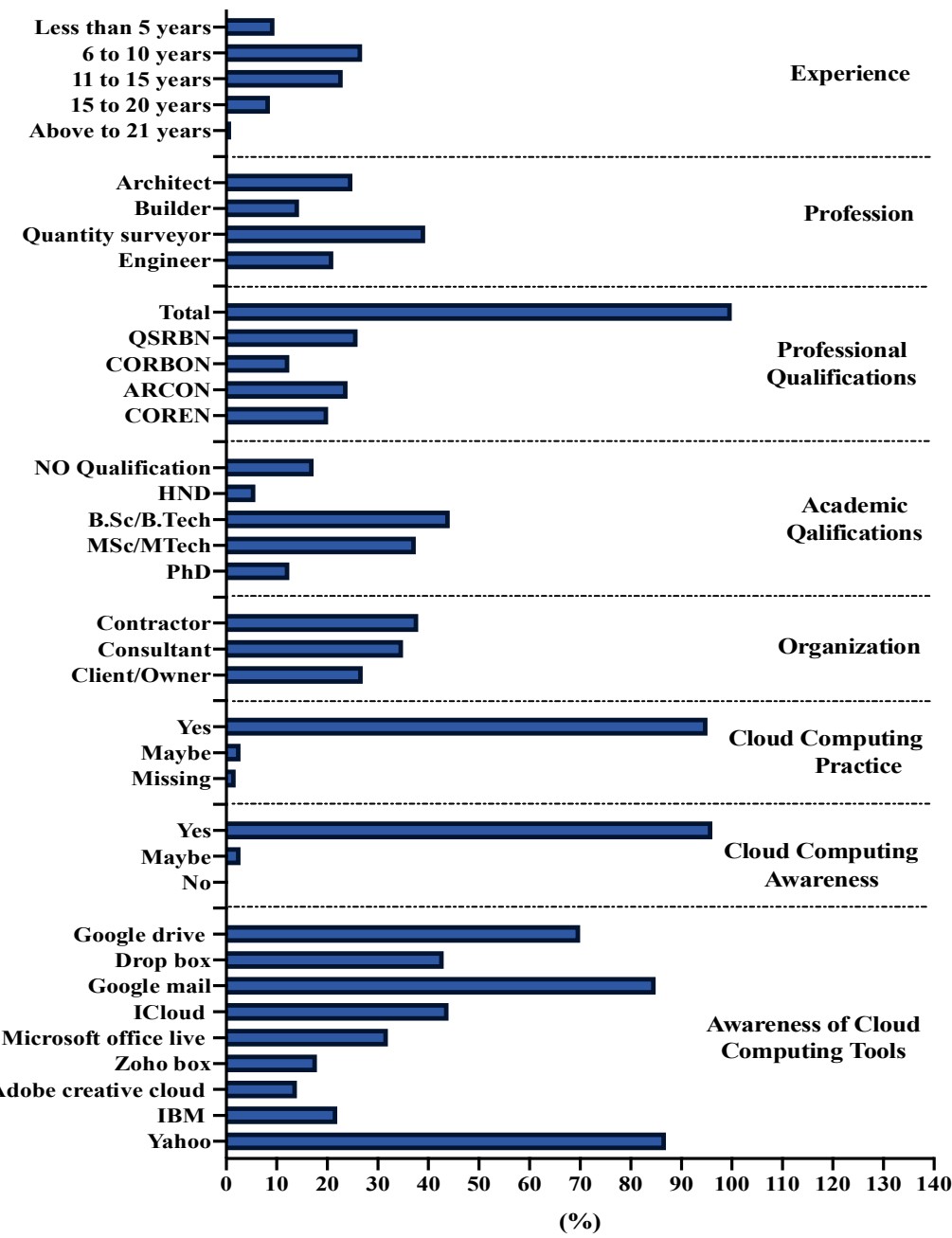

**Figure 4.** Demographic profiles.

*4.2. Classifying the Constructs of the Model*

Based on the Kaiser–Meyer–Olkin (KMO) sampling adequacy, Table 3 reveals that the data obtained were acceptable for factor analysis. The Kaiser–Mayer–Olkin (KMO = 0.676) sampling adequacy test shows that 67.6 percent of the data obtained was appropriate for the factor analysis (Table 3). Table 3 also displays that the *p*-value is less than 0.05, implying that the data is fit for EFA analysis with a freedom rate of 66 and an assessed chi-square of 784.50 for this data. However, Bartlett's test is significant (*p*-value = 0.000), implying that the association is an identity matrix. It also suggests that the correlation matrix of the above-listed items has a significant correlation of 5 percent, specifying the suitability of using EFA. In addition, as shown in Table 4, the principal component analysis (PCA) revealed the existence of four items with eigenvalues greater than one, accounting for (17.5%), (16.8%), (16.3%), (16%), and (15.6%), respectively, of the variance in the areas of cloud computing use in the construction industry. Nonetheless, the second item on the

screen plot indicated a distinct break. The number of factors the analysis should produce is indicated by the curve slope leveling off (Figure 5).

**Table 3.** Construction activities KMO and Bartlett's Test.

| KMO and Bartlett's Test | | |
|---|---|---|
| Kaiser–Meyer–Olkin Measure of Sampling Adequacy | | 0.676 |
| Bartlett's Test of Sphericity | Approx. Chi-Square | 784.50 |
| | Df | 66 |
| | Sig. | 0.000 |

**Table 4.** Construction activities total variance explained.

| Component | Initial Eigenvalues | | | Rotation Sums of Squared Loadings | | |
|---|---|---|---|---|---|---|
| | Total | % of Variance | Cumulative % | Total | % of Variance | Cumulative % |
| 1 | 4.05 | 33.71 | 33.71 | 2.10 | 17.50 | 17.50 |
| 2 | 2.20 | 18.34 | 52.04 | 2.03 | 16.89 | 34.39 |
| 3 | 1.47 | 12.27 | 64.31 | 1.96 | 16.31 | 50.69 |
| 4 | 1.11 | 9.27 | 73.58 | 1.91 | 15.92 | 66.62 |
| 5 | 1.04 | 8.63 | 82.21 | 1.87 | 15.59 | 82.21 |
| 6 | 0.67 | 5.56 | 87.77 | | | |
| 7 | 0.47 | 3.95 | 91.72 | | | |
| 8 | 0.37 | 3.07 | 94.79 | | | |
| 9 | 0.30 | 2.46 | 97.25 | | | |
| 10 | 0.19 | 1.60 | 98.84 | | | |
| 11 | 0.12 | 1.00 | 99.85 | | | |
| 12 | 0.02 | 0.15 | 100.00 | | | |

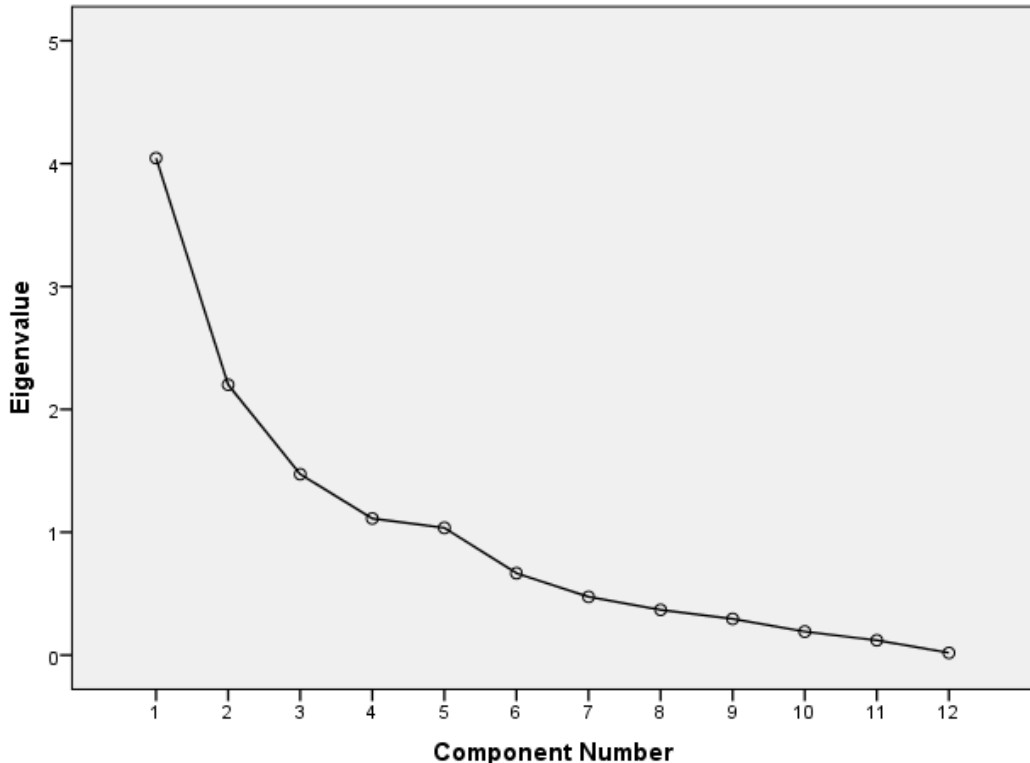

**Figure 5.** Screen plot result for the construction activities.

A model of five applications represents the cloud computing areas of application in the Nigerian construction industry. Table 5 shows that the factors are classified based on the varimax rotation, and each variable strongly influences each of the applications. Therefore, it is essential to identify these factors before interpreting the five extracted applications.

**Table 5.** Construction activities rotated component matrix.

| Activities | F1 | F2 | F3 | F4 | F5 |
|:---:|:---:|:---:|:---:|:---:|:---:|
| C1 | 0.08 | 0.05 | 0.87 | 0.08 | 0.14 |
| C2 | 0.22 | 0.10 | 0.82 | 0.00 | 0.27 |
| C3 | 0.13 | 0.01 | 0.36 | 0.15 | 0.79 |
| C9 | 0.08 | 0.98 | 0.01 | 0.08 | 0.11 |
| C10 | 0.09 | 0.98 | 0.06 | 0.04 | 0.10 |
| C4 | 0.47 | 0.04 | 0.15 | 0.10 | 0.61 |
| C5 | 0.01 | 0.26 | 0.10 | −0.02 | 0.75 |
| C6 | 0.70 | 0.12 | 0.51 | 0.18 | 0.05 |
| C7 | 0.89 | 0.08 | 0.26 | 0.11 | 0.01 |
| C8 | 0.70 | 0.10 | 0.14 | 0.12 | 0.50 |
| C11 | 0.11 | 0.05 | 0.00 | 0.96 | 0.08 |
| C912 | 0.15 | 0.06 | 0.12 | 0.94 | 0.07 |

The naming of these variables is arbitrary since the individual's experience and training determine it. There is no particular set method for naming the variables. As a result, carefully naming these variables was considered necessary for this study. Table 6 shows how each factor is labeled and explained as named in the table, i.e., "Pre-Contract Stage, Management, Design and Storage, Estimation ad Communications, and Finally, Back-office activities."

**Table 6.** Related components of the construction activities.

| Factor Component | Codes | Factor Loading |
|:---:|:---:|:---:|
| Pre-contract stage | C1 | 0.87 |
| | C2 | 0.82 |
| Design and storage | C3 | 0.79 |
| | C4 | 0.61 |
| | C5 | 0.75 |
| Management | C6 | 0.70 |
| | C7 | 0.89 |
| | C8 | 0.70 |
| Estimation and communication | C9 | 0.98 |
| | C10 | 0.98 |
| Back-office activities | C11 | 0.96 |
| | C12 | 0.94 |

*4.3. Common Method Bias*

A common method bias (CMB) is perhaps a unit of errors (variance) that measures the impacts of the validity of a study's findings and results. Therefore, the measured and predicted variables are subject to systematic error variances [129]. Harman's single-factor assessment of models can be used to evaluate the measured structures [130]. This study measured the standard method's variable using the single-factor test [131]. The CMB does not impact the results if the variables' total variance is less than 50 percent [130]. For example, as shown in Table 4, one set of variables accounts for 21.72 percent of the variance, indicating that the CMV cannot affect the findings since it is less than 50 percent [130].

### 4.4. Measurement Model (First-Order Construct)

Figure 6 shows the SEM results for the study's conceptual model (from Figure 2). Each model's cloud computing and construction activities construct were itemized and categorized in Tables 1 and 2 based on the items from previous studies. For every reflective construct, the indicators should be in harmony with each construct [132]. In general, outer loading indicators of 0.40 to 0.70 should be considered for deletion only if the removal significantly improves the composites' reliability and the AVE [118]. The outer loadings for all variables in the original and modified measurement models are shown in Table 7 and Figure 6. As a result, all extreme loads were excluded from the initial measurement model, except for one item related to human satisfaction "D1," which was removed due to a low loading factor of less than 0.65, confirming their low impact on the related constructs. After removing these variables, a modified model was tested further. Hair Jr, et al. [133] recommended evaluating the internal consistency of composite reliability (cr) using Cronbach alpha limits, which is the sensitivity to the number of elements involved as recommended. Hair Jr, et al. [133] recommend values above 0.70 for any study and above 0.60 for exploratory analysis [134]. As shown in Table 5, all models met the cr > 0.70 thresholds and were accepted. As indicated by Wong [134], AVE is a standard measure for assessing the convergent validity of constructs in the model, with values above 0.50 implying a sufficient convergent value. Furthermore, Table 7 shows that every single construct passes this test. This means that the measurement variables absorb at least half of the measurement variation [135].

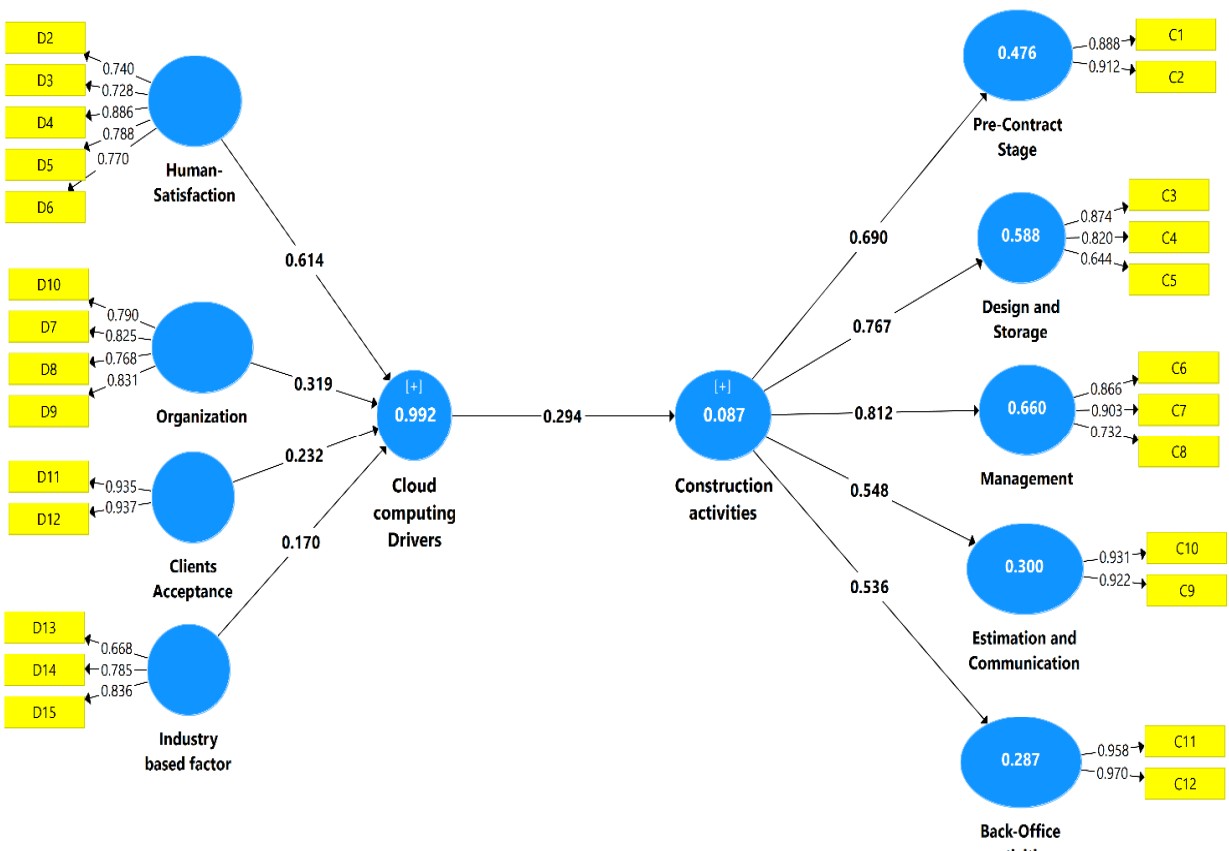

**Figure 6.** Structural model with path coefficient and $R^2$ values.

**Table 7.** Convergint validity results.

| Constructs | Item | Outer Loading | | Cronbach's Alpha | Composite Reliability | AVE |
|---|---|---|---|---|---|---|
| | | Initial | Modified | | | |
| Human satisfaction | D.1 | 0.549 | deleted | 0.842 | 0.888 | 0.615 |
| | D.2 | 0.751 | 0.740 | | | |
| | D.3 | 0.716 | 0.728 | | | |
| | D.4 | 0.884 | 0.886 | | | |
| | D.5 | 0.768 | 0.788 | | | |
| | D.6 | 0.760 | 0.770 | | | |
| Organization factor | D.7 | 0.827 | 0.825 | 0.818 | 0.879 | 0.646 |
| | D.8 | 0.768 | 0.768 | | | |
| | D.9 | 0.833 | 0.831 | | | |
| | D.10 | 0.786 | 0.790 | | | |
| Clients acceptance | D.11 | 0.936 | 0.935 | 0.858 | 0.934 | 0.875 |
| | D.12 | 0.935 | 0.937 | | | |
| Industry-based factor | D.13 | 0.702 | 0.668 | 0.659 | 0.809 | 0.587 |
| | D.14 | 0.785 | 0.785 | | | |
| | D.15 | 0.814 | 0.836 | | | |
| Pre-contract stage | C1 | 0.887 | 0.888 | 0.766 | 0.895 | 0.81 |
| | C2 | 0.912 | 0.912 | | | |
| Design and storage | C3 | 0.874 | 0.874 | 0.687 | 0.826 | 0.617 |
| | C4 | 0.820 | 0.820 | | | |
| | C5 | 0.644 | 0.644 | | | |
| Management | C6 | 0.866 | 0.866 | 0.782 | 0.874 | 0.701 |
| | C7 | 0.903 | 0.903 | | | |
| | C8 | 0.732 | 0.732 | | | |
| Estimation and communication | C9 | 0.922 | 0.922 | 0.834 | 0.924 | 0.858 |
| | C10 | 0.930 | 0.931 | | | |
| Back-office activities | C11 | 0.958 | 0.958 | 0.925 | 0.964 | 0.93 |

Discriminant validity is well-defined if the construct varies appropriately from the other constructs based on the criteria found. Consequently, the construct's founding discriminative validity becomes unique and describes the phenomena that other frameworks in the paradigm cannot ascertain [121]. Fornell and Larcker's (1981) criteria, the HTMT (Hetrotrait–Monotrait ratio of correlations) criterion, and the cross-loading criterion may all be used to evaluate bias in its validity. The HTMT criterion can be used to assess discrimination. The square root of each construct's AVE can be matched to the coefficients of one construct with every other construct to determine its discriminative validity. According to Fornell and Larcker [136] rules, the correlations between the latent variables must be more significant than the square root of the AVE. Table 8 shows that the result supports the measurement model's discriminant validity [137].

Nevertheless, many authors, such as Fornell and Larcker [136], opposed the idea of the characteristic discriminatory validity as a criterion. As a result, Henseler, et al. [138] proposed a different form of discriminatory validity assessment, namely the hetrotrait–monotrait Ratio (HTMT). "HTMT" is a modern method of evaluating the discriminative validity of SEMs based on the variance and approximating the specific link between the two constructs when accurately calculated (i.e., whether they are error-free and reliable). The HTMT model has also been used in this analysis to evaluate the validity of the discriminant. Hair, et al. [139] suggested a lower HTMT value of 0.85 and 0.90, which means that both constructs were separate. If the model's constructs are conceptually identical, the HTMT value of 0.90 will be lower than 0.85 if they are conceptually unique. Table 9 shows the HTMT results for all of the components in this study. Therefore, enough discriminative validity has been demonstrated in the construct.

**Table 8.** Fornell–Larcker results.

| Constructs | Back-Office Activities | Clients Acceptance | Design and Storage | Estimation and Communication | Human-Satisfaction | Industry-Based Factor | Management | Organization | Pre-Contract Stage |
|---|---|---|---|---|---|---|---|---|---|
| Back-office activities | 0.964 | - | - | - | - | - | - | - | - |
| Clients' acceptance | 0.287 | 0.936 | - | - | - | - | - | - | - |
| Design and storage | 0.24 | 0.211 | 0.785 | - | - | - | - | - | - |
| Estimation | 0.294 | 0.244 | 0.262 | 0.926 | - | - | - | - | - |
| Human satisfaction | 0.028 | 0.33 | 0.244 | 0.155 | 0.785 | - | - | - | - |
| Industry-based factor | 0.217 | 0.404 | 0.314 | 0.153 | 0.331 | 0.766 | - | - | - |
| Management | 0.323 | 0.051 | 0.492 | 0.279 | 0.189 | 0.19 | 0.837 | - | - |
| Organization | 0.087 | 0.254 | 0.3 | 0.124 | 0.407 | 0.355 | 0.034 | 0.804 |  |
| Pre-contract stage | 0.159 | 0.059 | 0.476 | 0.23 | 0.124 | −0.004 | 0.468 | 0.042 | 0.9 |

**Table 9.** HTMT Value.

| Constructs | Back-Office Activities | Clients Acceptance | Design and Storage | Estimation and Communication | Human-Satisfaction | Industry-Based Factor | Management | Organization | Pre-Contract Stage |
|---|---|---|---|---|---|---|---|---|---|
| Back-office activities |  | - | - | - | - | - | - | - | - |
| Clients acceptance | 0.32 | - | - | - | - | - | - | - | - |
| Design and storage | 0.279 | 0.26 | - | - | - | - | - | - | - |
| Estimation and communication | 0.333 | 0.29 | 0.334 | - | - | - | - | - | - |
| Human satisfaction | 0.112 | 0.379 | 0.346 | 0.207 | - | - | - | - | - |
| Industry-based factor | 0.305 | 0.541 | 0.455 | 0.212 | 0.406 |  | - | - | - |
| Management | 0.376 | 0.111 | 0.665 | 0.343 | 0.231 | 0.348 | - | - | - |
| Organization | 0.131 | 0.286 | 0.399 | 0.194 | 0.469 | 0.513 | 0.18 | - | - |
| Pre-contract Stage | 0.186 | 0.073 | 0.637 | 0.286 | 0.211 | 0.193 | 0.583 | 0.129 | - |

The third method shows that the cross-loading criterion was applied to evaluate discriminative validity throughout this study. It means that the construct's loading indicators (items) must be greater than other items' loading. It aimed to assess the indicator loading on a latent structure and be more excellent in each row and other constructs. As seen in Table 10, the findings demonstrate that almost all latent construct indicators are loaded higher than the cross-load of other constructs in the row. Furthermore, the outcome revealed a significant amount of one-dimensionality for each construct.

**Table 10.** Cross loadings results.

| Drivers | Human Satisfaction | Organization | Clients Acceptance | Industry | Pre-Contract | Design and Storage | Management | Estimation and Communication | Back-Office Activities |
|---|---|---|---|---|---|---|---|---|---|
| D2 | 0.74 | 0.233 | 0.184 | 0.279 | 0 | 0.112 | 0.055 | 0.034 | 0.155 |
| D3 | 0.728 | 0.291 | 0.18 | 0.114 | 0.017 | 0.173 | 0.231 | 0.056 | 0.059 |
| D4 | 0.886 | 0.416 | 0.316 | 0.258 | 0.086 | 0.207 | 0.189 | 0.183 | 0.09 |
| D5 | 0.788 | 0.311 | 0.404 | 0.36 | 0.202 | 0.254 | 0.165 | 0.185 | 0.048 |
| D6 | 0.77 | 0.325 | 0.178 | 0.272 | 0.193 | 0.2 | 0.094 | 0.222 | 0.031 |
| D7 | 0.218 | 0.825 | 0.063 | 0.246 | 0.008 | 0.257 | 0.06 | 0.036 | 0.014 |
| D8 | 0.316 | 0.768 | 0.183 | 0.277 | 0.01 | 0.154 | 0.037 | 0.115 | 0.069 |
| D9 | 0.304 | 0.831 | 0.231 | 0.288 | 0.07 | 0.177 | 0.144 | 0.082 | 0.07 |
| D10 | 0.427 | 0.79 | 0.295 | 0.314 | 0.174 | 0.353 | 0.203 | 0.284 | 0.128 |
| D11 | 0.287 | 0.217 | 0.935 | 0.468 | 0.045 | 0.187 | 0.047 | 0.23 | 0.271 |
| D12 | 0.329 | 0.258 | 0.937 | 0.288 | 0.065 | 0.208 | 0.049 | 0.226 | 0.265 |
| D13 | 0.055 | 0.462 | 0.308 | 0.668 | 0.188 | 0.164 | 0.146 | 0.117 | 0.256 |
| D14 | 0.213 | 0.18 | 0.312 | 0.785 | 0.07 | 0.329 | 0.247 | 0.136 | 0.159 |
| D15 | 0.405 | 0.237 | 0.319 | 0.836 | 0.051 | 0.23 | 0.08 | 0.108 | 0.125 |
| C1 | 0.047 | 0.013 | 0.059 | 0.003 | 0.888 | 0.373 | 0.357 | 0.213 | 0.186 |
| C2 | 0.17 | 0.059 | 0.048 | 0.01 | 0.912 | 0.478 | 0.478 | 0.201 | 0.105 |
| C3 | 0.191 | 0.177 | 0.124 | 0.222 | 0.493 | 0.874 | 0.386 | 0.196 | 0.231 |
| C4 | 0.13 | 0.296 | 0.279 | 0.326 | 0.343 | 0.82 | 0.483 | 0.277 | 0.236 |
| C5 | 0.298 | 0.247 | 0.067 | 0.175 | 0.26 | 0.644 | 0.262 | 0.123 | 0.06 |
| C6 | 0.193 | 0.036 | 0.042 | 0.035 | 0.554 | 0.408 | 0.866 | 0.25 | 0.293 |
| C7 | 0.13 | 0.011 | 0.083 | 0.088 | 0.393 | 0.362 | 0.903 | 0.258 | 0.262 |
| C8 | 0.148 | 0.041 | 0.102 | 0.398 | 0.193 | 0.479 | 0.732 | 0.186 | 0.254 |
| C9 | 0.126 | 0.132 | 0.268 | 0.156 | 0.15 | 0.227 | 0.261 | 0.922 | 0.302 |
| C10 | 0.159 | 0.098 | 0.186 | 0.129 | 0.272 | 0.257 | 0.256 | 0.931 | 0.244 |
| C11 | 0.059 | 0.131 | 0.251 | 0.26 | 0.093 | 0.214 | 0.267 | 0.255 | 0.958 |
| C12 | 0.023 | 0.044 | 0.298 | 0.166 | 0.205 | 0.246 | 0.348 | 0.308 | 0.97 |

*4.5. Measurement Model (Second-Order Construct)*

The key variables were second-order latent variables. Therefore, the bootstrap method is required as a critical aspect of all first-order latent variables and was assessed. However, one of the constructs of cloud computing application drivers was constructive, while the project construction activities were reflective.

A forecast of a significant correlation between the formative measurement model indicators is usually not indicated. In addition, a high correlation in the formative elements indicates collinearity, labeled as disorderly [121]. Hence, we investigate the construct's collinearity formative elements by examining the variable inflation factor's (VIF) value. When dealing with a reflective-formative second-order construct in the analyses, we used internal VIF values to check for collinearity. A robust standard path coefficient was found β (outer weight) for four first-order subscales for cloud computing drivers, including human satisfaction, organization, client acceptance, and industry-based factors. As shown in Table 11, the overall external loading was observed for human satisfaction (β = 0.614, $p < 0.001$), followed by organization (β = 0.319, $p < 0.001$), clients acceptance (β = 0.232, $p < 0.001$), and industry-based factor (β = 0.17, $p < 0.001$). The VIF values for these figures were less than 3.5, showing that these subdomains contribute to the higher-order construct separately.

**Table 11.** Test of second-order models using bootstrapping for formative construct.

| Path | $\beta$ | SE | t Values | p Values | VIF |
|---|---|---|---|---|---|
| Human satisfaction -> Cloud computing drivers | 0.614 | 0.054 | 11.312 | <0.001 | 1.309 |
| Industry-based factor -> Cloud computing drivers | 0.17 | 0.041 | 4.183 | <0.001 | 1.328 |
| Organization -> Cloud computing drivers | 0.319 | 0.063 | 5.096 | <0.001 | 1.286 |
| Clients acceptance -> Cloud computing drivers | 0.232 | 0.047 | 4.915 | <0.001 | 1.264 |

The second-order framework expressed in this model indicates the project performance outcomes from five subcomponents: back office, architecture and storage, estimate and communications, management, and the pre-contract stage. Hence, all five components should be added as latent second-order variables to the construction activities. The standardized path coefficient (outer loads) was considered necessary in the second-order latent variables as it is above 0.5, as shown in Table 12.

**Table 12.** Test of reflective second-order models.

| Path | $\beta$ | SE | t Values | p Values |
|---|---|---|---|---|
| Construction activities -> Back-office activities | 0.536 | 0.123 | 4.359 | 0 |
| Construction activities -> Design and storage | 0.767 | 0.055 | 13.997 | 0 |
| Construction activities -> Estimation | 0.548 | 0.094 | 5.853 | 0 |
| Construction activities -> Management | 0.812 | 0.034 | 23.795 | 0 |
| Construction activities -> Pre-contract stage | 0.69 | 0.074 | 9.372 | 0 |

*4.6. Structural Model*

Path analysis is a linear regression statistics technique. It is also a popular approach for examining all dynamic relationships simultaneously [140]. The SEM primarily focuses on the overall model fit, with assumed scale, path, and significance parameter assessments [141]. "The value among each path illustrates the path coefficient, which evaluates the level of impact of a construct on another" [142]. Finally, the analysis relationship was confirmed based on the tested hypotheses outlined in Figure 2 [143].

PLS-SEM analyzed cloud computing drivers' impact on construction activities as presented in Figure 7 of the research hypothesis model. The significance of the assumption in the bootstrapping technique was evaluated based on the model. The primary dataset of random re-sampling involves the bootstrapping process to produce a new sample size as the original dataset. This approach tests the reliability and statistical validity; hence, the path coefficients' error was measured [143]. The standardized path coefficients ($\beta$), the *p*-values, and the pathway significance $R^2$ were evaluated for every endogenous construct. Table 13 and Figure 7 show the outcome of bootstrapping, whereby *p*-values for each path are shown. The findings ($\beta = 0.294$, $p = 0.007$) show a statistically positive and significant effect on construction activities. In addition, the $R^2$ value was used to adjust the project progress as the dependent variable (0.078) in the proposed model. It indicated that 0.7 percent of the projects' development works could be classified as an exogenous latent variable (cloud computing drivers) [144]. Chin [143] suggested that such results indicate that the cloud drivers' size was small.

**Table 13.** Relative path for the model.

| Path | $\beta$ | SE | t Value | p Value |
|---|---|---|---|---|
| Cloud computing drivers -> Construction activities | 0.294 | 0.109 | 2.688 | 0.007 |

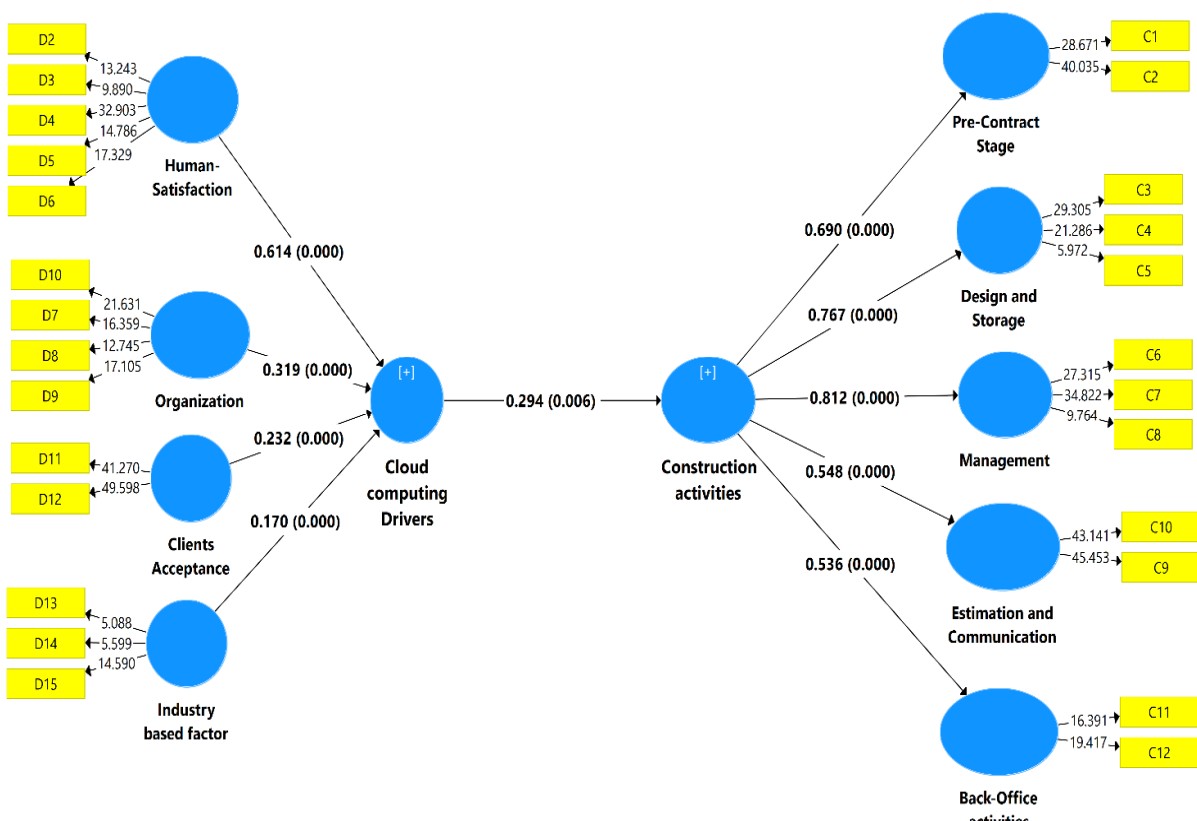

**Figure 7.** Bootstrapping analysis.

*4.7. The Importance-Performance Matrix Analysis (IPMA)*

In defining the dependent variable in the path model, PLS-SEM proves the relative importance value of an independent variable [133,145]. At the same time, importance-performance matrix analysis (IPMA) enhances the PLS-SEM findings by considering each performance variable output. Thus, the findings can be concluded from two crucial aspects of management decisions: importance and performance [133]. The overall impacts of the structural model (significance) and the variable mean latent value (score) of the scales are used to stress the significant fields of management optimization (or the particular emphasis of the model). Therefore, in this research, IPMA was employed as a cloud-based variable. The result as shown in Table 14 displays the exogenous variable's degree of importance and performance (cloud computing drivers).

**Table 14.** Importance and Total Effects for the IPMA Cloud computing drivers.

| Predictor | Importance | Performances |
|---|---|---|
| Cloud computing drivers | 0.313 | 63.449 |

## 5. Discussion

Many industrial and daily activities in built environments rely on robust indoor positioning services. Customers' interest in internet-based real-time services has grown in recent years, thanks partly to the quick, convenient, and cost-effective delivery of services [146]. Cloud computing is one such service [147]. The introduction of cloud computing signifies a major shift in how information technology (IT) services are conceived, developed, implemented, scaled, upgraded, managed, and paid for [61]. "Cloud computing has greatly influenced the IT industry and attracted much attention from IS scholars" [147]. Many businesses have set up cloud centers to provide different services (e.g., infrastructure as a service) to businesses and supply chain partners on a pay-as-you-go basis [148]. Fur-

thermore, the most recent technological innovation has made cloud-based distribution the principal method for digital content delivery (e.g., documents, mails, videos, images, music, and gaming) [149] in communications, information and management control, construction project management, organizational optimization performance, and enriched decision-making. Therefore, cloud-computing adoption is essential, especially in the construction industry [86].

*5.1. Identified Level of Success in Construction Activities*

The findings indicate that management activities were ranked first, with an outer loading score of 0.810. The results are concurrent with Muhammad Abedi [82] findings, which showed that cloud computing helps clients (commercial enterprises and members of society) view, share, and disseminate data, apps, and resources on different servers. Additionally, it encourages clients to use other applications without installation concerns, helping users view their varied data over the Web from their smartphone or computer. The time measured throughout the user value chain is responsible for the project's success and shows that cloud computing may save project time. The time factor is usually essential in every project work as an opportunity to schedule activities effectively, including time for the project's design, the time required to execute variation orders, time to access services over the project period, and time necessary to obtain authority's approval [147]. Thus, projects are considered successful if finished within the specified project time.

Design and storage are the second factors, with a 0.767 external loading. This value is similar to Afolabi [53] findings that the construction industries' cloud computing application areas are structural analysis design, architectural design, and data storage. These findings further concurred with Oliveira, et al. [79]. Notably, the cloud computing paradigm allows access to all facets of software creation (from design to testing, version management, servicing, and hosting) via the internet. Furthermore, through software as service (SaaS), clients can access the centrally hosted programs in the cloud via different options (such as a web browser or smartphone application) rather than running the software on their personal computers. Kumar, et al. [150] claimed that cloud computing could be applicable in different ways to the construction industry, namely modeling, structural analysis, cost estimation, planning and monitoring, and procurement.

Pre-contract activities are the third-ranked factor, with an outer load of 0.68. It concurred with Afolabi's [53] assertion that pre-contract work would fruitfully use cloud computing experience in contract documentation and project planning. Among the fundamental principles in executing construction projects are generally implemented during pre-construction. Construction sites deal with much information before construction operations are conducted. Construction workers require vast information to help with construction operations, from project design drafts to personal data documentation.

Estimation and communications activities were ranked as the fourth outer loading factor with 0.548, similar to Afolabi's [53] findings on the importance of cloud computing practices in the construction industry on cost and communication. Afolabi [53] has shown that several construction industry shareholders use their Gmail or Yahoo mail to access the cloud. Cloud computing's critical potential is the speed with which data is shared between experts and other construction industry players. In addition to the traditional methodologies and procedures, Muhammad Abedi [82] argued that cloud computing deployment would lead to collaboration and coordination among partners and increase operating efficiency in the construction industry. Implementing cloud computing allows for more flexibility in developing a successful strategy regarding income generation, efficiency, and effectiveness. Furthermore, when nonmonetary advantages are taken into account, cloud computing costs are reduced [151–153]

Back-office activities were ranked as the last factor, with an outer loading of 0.537. The result agrees with Tehrani and Shirazi [24], Muhammad Abedi [82]'s findings on the gains of cloud computing tools. Therefore, construction experts believe that the construction

industry's cloud computing practices can be achieved using back-office activities with significant advantages. This claim is also followed by Tehrani and Shirazi [24], Afolabi [53].

## 5.2. Identified Impact of Cloud Computing on Construction Activities

The construction industry is a diverse and dynamic sector. Construction projects entail large-scale activities and interrelated operations that significantly impact the environment, social activities, and the economy over the project life cycle [25]. Therefore, it is crucial to measure cloud computing deployment drivers' effect on construction activities and assess their impacts to improve construction efficiency. The dependent and independent variables' assessment was utilized to determine its influence on construction work by installing cloud computing drivers. The results indicate that 0.087% contributes to the project's successful performance by implementing cloud computing drivers. Implementing the cloud computing driver was also significantly linked to construction activity. The significance of $\beta = 0.299$ is vital if the company or organization found 1 unit of the cloud computing driver. Thus, this increases the performance of construction activities by 0.299%. The results also showed that a few of the cloud driver's outcomes would allow construction stakeholders to efficiently handle the project and fulfill the consumer demands on quality and prices.

Based on these findings, it can be inferred that the cloud driver's contribution would influence construction activities by evaluating the defined cloud computing application effectiveness in project management. The results satisfied users' expectations on its success in construction activities. The objective herein is accomplished. It is analogous to earlier studies that showed that resources, such as time and labor usually dedicated to the company's IT department, can be invested in other crucial fields. Small and medium companies whose primary domain is not IT do not have to manage or upgrade their information system (IS); instead, they should concentrate on their core industrial vision to improve their companies' success and productivity. Afolabi [53] sees cloud computing as helping companies achieve their growth goals by saving energy, making them more efficient, improving consistency and efficiency, and concentrating on their core activities instead of non-core activities like technology management upgrades. It motivates employees to be more innovative and discover new techniques to do business [24]. Additionally, there is perceived usefulness to cloud use, which increases the availability of resources, and reduces hardware infrastructure expenses, accessibility, collaboration opportunities, economic reductions, productivity improvements, and administrative control [154,155].

The costs of cloud technologies are other aspects that discourage IT resource users from procuring them [54]. It is undoubtedly accurate that cloud users could benefit from the pay-per-use model based on its scalability to harness its total value. Small and medium-sized companies in sub-Saharan Africa face several issues that require minimizing the capital costs of developing IT infrastructure systems [25]. Cloud computing applications reduce operational costs, personnel charges, and licensing fees for small and medium-sized companies to purchase their software from tech vendors. The cost of retaining massive energy-based computing facilities is also partially minimized [84], which will be an added advantage due to the unstable nature of energy in many African countries. Cloud computing requires an isolated location to run efficiently due to various machines' connections, independent from the software and rendering the local hardware used in cloud computing technology worthwhile [25]. Cloud computing is more adaptable than other computer technologies. It is more robust than other new technologies in the construction industry. Unlike other conventional storage methods, cloud computing allows users to connect and use the technology from anywhere they can access the network [24].

## 5.3. Drivers of Cloud Computing Implementation Framework for Achieving Construction Success

Construction success can be achieved through cloud computing (Figure 8). According to the proposed PLS-SEM model, there is a correlation between cloud computing driver's implementation and construction activities. Hence, structural models of CC application

drivers confirm the criteria for deploying cloud computing [154]. In addition, this technique is applied in the domain of system development and construction [38,155]. Most stakeholders agree that these drivers are critical for project improvement [38,156]. Utilizing these drivers in any domain can be helpful in decision-making [157].

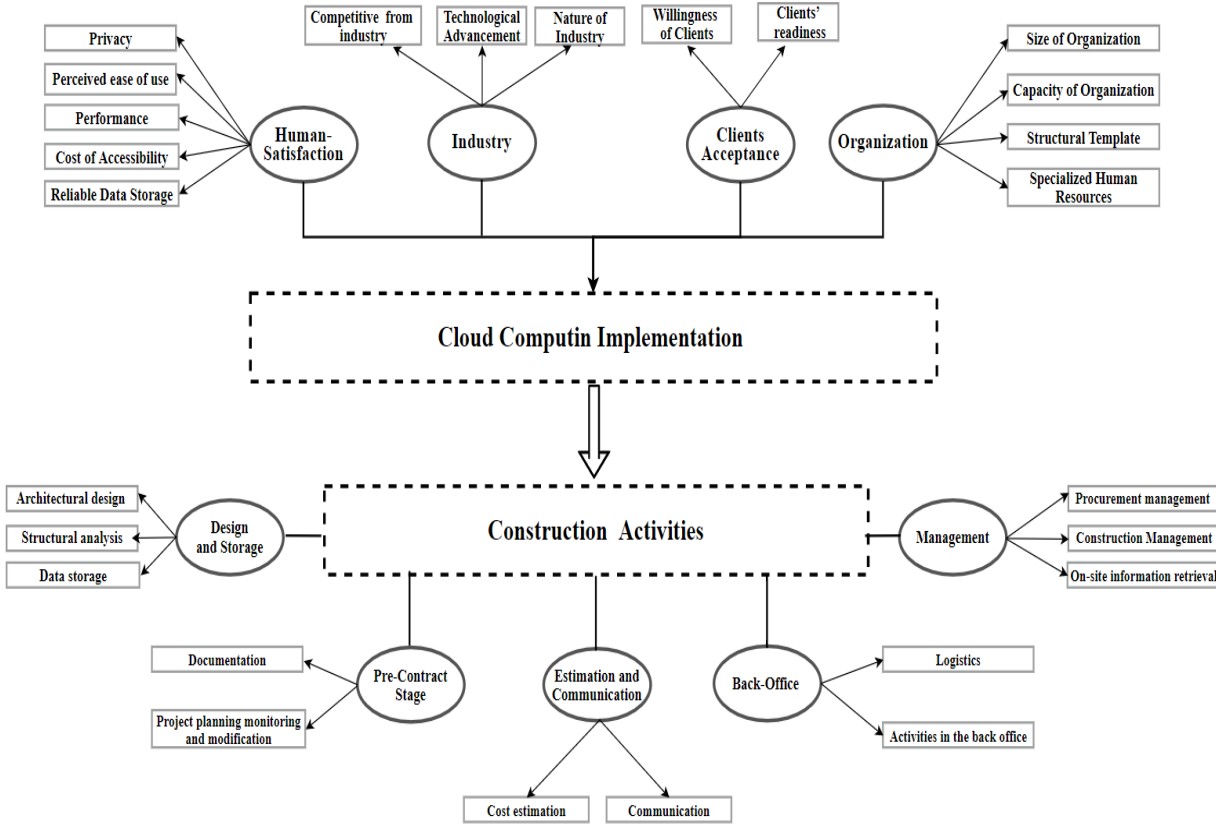

**Figure 8.** Drivers for cloud computing implementation for achieving construction success.

The suggested framework explains and links the various drivers [156]. According to the "framework," a causal connection is a graph that shows the variables in an abstract model. This graph would form the basis for the conceptions in the diagram [156]. The suggested framework views the issue as a logical problem. It controls the concept-related dimension (cloud computing drivers and construction activities) resulting from the suggested PLS-SEM. A framework composed of assertions linking the abstract ideas to empirical data in a theoretical form was suggested by Leshem and Trafford [157]. The proposed framework accommodated abstract events of similar situations [157]. They create a connection between theory and reality that many scholars miss. As a result, the proposed framework includes a database of CC standards and the variables that influence their competitiveness and worldwide market survival via cloud computing integration. These CC drivers should be identified before the CC can be effectively adopted in the Nigerian construction sector, necessitating further policy consideration. The framework elements created by the proposed model are shown in the subsequent sections.

5.3.1. Human Satisfaction

"The importance of stakeholders in construction projects is undeniable" [158]. Human satisfaction for stakeholders is unquestionably essential in cloud computing adoption in building projects, as shown by many studies. According to the PLS-SEM model, this element has the most impact on the drivers of CC deployment, with an external coefficient of 0.614 on the "Human-Satisfaction" component, signifying the most significant influence. Its first principal component comprised drivers such as availability, privacy, perceived

ease-of-use, performance, cost of accessibility, and reliability of data storage. According to Gangwar, et al. [50], performance and availability drive CC adoption. Additionally, a cloud computing study conducted by Gaurangkumar and Minubhai [159] found a pressing need to provide solutions that can build trust and happiness among users to quickly adopt cloud computing technology in their construction activities.

### 5.3.2. Organization

In terms of the second principal component, it is concerned with "organization." It includes drivers such as "Organization size, Organization capacity, Structural template, and Specialized Human Resources," among other things. When applied to drivers, the "workshop dynamics" path coefficient equals 0.319. This value was similar to Simamora and Sarmedy [160]. Conversely, Priyadarshinee [161] stated that the company's cloud computing efficiency could depend on the firm's use of the cloud delivery strategy. Organizations need to adopt cloud computing since the firm size and structure template are required for cloud computing adoption, and larger companies have the resources to assist in implementing cloud computing [50,79].

### 5.3.3. Clients' Acceptance

"Clients' Acceptance" is the third major component of the overall process. Tehrani and Shirazi [24] showed that top management (client) support and the willingness of stakeholders to participate in cloud computing adoption in the construction industry have a significant impact on the adoption of cloud computing. This item accounts for a 0.232 path coefficient and is composed of "the willingness of clients" and "Client's readiness." Some construction stakeholders believe these factors impact clients' costs to adopt cloud computing services. Murad and Fatema [162] pointed out that cost is an essential issue that must be considered when determining whether or not cloud computing services should be used. In addition, the responsibility (i.e., mandatory environmental rules) for introducing sustainability standards will be addressed if the authorities collaborate with the client and top management to identify and provide proper support (i.e., financial incentives) to meet those criteria. Thus, top management impacts the implementation of cloud technology in an organization because it may connect people with advances in cloud services, adding more value to the company's vision [163]. The ICT division, particularly in cloud computing, may flourish with the backing of top management. Consequently, the company's operations will align more with its vision and mission since cloud computing will support those activities [164]. Thus, improvements will be achieved in the processes.

### 5.3.4. Industry-Based Factor

The "Industry-based factor" is the last component of the driver's factors needed for adopting cloud computing. A 0.17-to-one external coefficient characterizes it. It includes the following drivers: "Nature of the Industry, Competitive Pressure from within the industry, and technological advancement." In support of the claim made by Oliveira, et al. [79] that advancement of technology, increased competition from within the industry, as well as the nature of the industry each have a substantial influence on the deployment of cloud computing in the construction sector and increases innovation throughout the life cycle of the project, as innovation is among the most significant entities that the construction industry must retain for resource management [165].

## 6. Conclusions and Implications

Nigeria is categorized as a densely populated developing nation with poor environmental concerns in the nation's context. On the other hand, it is supposed to be a sustainable, stable, and diversified nation. Furthermore, Nigeria's construction industry lacks research on cloud computing implementation drivers for construction activities. Thus, these findings provide the basis for implementing cloud computing in the construction industry. Results can also be used in the Nigerian construction industry as a base for cloud

computing implementation. It would become a worthwhile policy that minimizes project costs by allocating them via cloud computing technology.

Cloud computing is highly dependent on construction activities, and it is very moderate in developing nations. Nigeria has encountered irregularities and inconsistencies in construction quality in large-scale projects. Cloud computing can be adopted to mitigate such problems. The EFA analysis was conducted to categorize the construction activities in the Nigerian construction industry. The outcomes from EFA illustrate that these activities can be categorized under five main components: pre-contract stage, management, design and storage, estimation and communications, and finally, back-office activities.

Furthermore, a PLS-SEM methodology was adopted to verify the relationship between cloud drivers' adoption and construction activities construct. Based on the data collected from 104 construction project experts, the structural model has established one direct path and nine indirect paths. It also verified the relationship between direct and indirect factors by linking the driver items with the variables. The findings of the developed model revealed that the human satisfaction driver is the most critical driver affecting the implementation of cloud computing, followed by "the Organization, Client Acceptance, and industry-based factor drivers" in that order of importance. The result obtained from the PLS-SEM also shows that cloud computing has the highest effect on management activities, followed by design and storage, pre-contract activities, estimation and communications, and back-office activities. The suggested model has shown that the findings have been accepted on the possibility of enhancing construction activity through implementing the identified cloud drivers.

Cloud computing drivers have also shown a significant influence on construction performance. Nevertheless, implementing cloud computing drivers has impacted construction activities' success and could lead to overall project success. Consequently, senior management can monitor their cloud computing resources and teams based on the cloud computing drivers' (constructs) effect and enhance their involvement by aiming to attain superior construction efficiency. Thus, this study generates vital theoretical contributions and managerial implications to the construction industry, which are given below.

## 7. Theoretical Contributions

For scholars, previous studies show a gap in cloud computing driver implementation. Hence, this study covered these gaps by examining the relationship between cloud computing drivers' implementation and construction activities contributing to the literature.

This research analyzed the drivers of cloud computing implementation via a predicted model. The suggested model establishes a prerequisite for cloud computing adoption, particularly in the Nigerian construction industry and other developing nations. The identified drivers would help overcome the present barriers hindering the implementation of cloud computing throughout the Nigerian construction industry. As a result, the gap in cloud computing theory and practices has been narrowed. The following part illustrates the finding's impact on the scientific body of knowledge:

- First, this research contributes to construction engineering management's knowledge body by giving an insight into cloud computing drivers and implementation guides. Thus, findings serve as a basis for academicians who want to research cloud technology drivers in emerging economies like Nigeria, especially in construction project management.
- Secondly, the methodological framework provides researchers with a foundation on cloud computing drivers' positive and substantial impact on construction activities. It also provides the basis for further research on its application in construction projects.
- Thirdly, it makes a conceptual contribution by identifying and defining additional constructs to the conceptual framework, such as the impact of cloud computing implementation drivers on construction activities.
- Fourthly, the range of construction-based cloud computing and implementation studies focused primarily on developed nations like the (United States, United Kingdom,

Hong Kong, and Australia) and other developing countries such as Malaysia, China, and Saudi Arabia. Therefore, few studies were conducted on implementing cloud computing in developing nations like Nigeria. Hence, it lays a promising opportunity for exploring cloud computing in developing countries by looking at the Nigerian construction industry's research context to improve the sustainability of local construction projects and bridge the knowledge gap.

- Fifthly, the output offers, for the first time, an effective prediction tool (PLS-SEM) to discuss the impact of cloud computing drivers on construction activities. As a result, this technology can boost conventional cloud computing adoption within the construction sector, especially in developing nations. This contribution is empirical as it tests a theoretical linkage between two constructs: the "cloud computing implementation drivers" and "construction" activities. Consequently, this research provides a mathematical framework for determining the cloud computing drivers that may be effectively employed in Nigeria and other developing nations, thus, assisting policymakers in expediting cloud computing adoption. The four elements of the cloud computing drivers in Nigeria's construction industry were analyzed using the PLS-SEM technique.

- Sixthly, it is evident from the results mentioned above that there is an increase in cloud computing awareness in the Nigerian construction industry (92%), which is anticipated to rise significantly within the next few years. Thus, this study provides evidence that cloud computing drivers have a vital and positive impact on construction activities. Consequently, this can encourage the Nigerian government and other local organizations to adopt cloud computing. Furthermore, such research will improve cloud computing adoption in developing nations. Therefore, the study makes significant contributions by adding new knowledge in a previously unexplored context of developing countries.

## 8. Managerial Implications

The reordering of cloud computing drivers could be beneficial in establishing a standard guide for stakeholders, such as project construction parties, by utilizing cloud computing to achieve a more successful construction process in their projects. In addition, the reordering may serve as a model for effectively developing a new framework integrating cloud computing construction players. However, this research makes a significant contribution to making managerial decisions, all of which have substantial implications in the construction industry in the following ways:

- Firstly, it provides a cloud computing implementation database and its related implications on construction activities—it highlights its competitive advantage and world market sustainability through several cloud computing integrations.

- Secondly, it allows owners, consulting firms, and contractors to analyze and decide on the best adoption driver of cloud computing models to maximize construction projects' planning, efficiency, and success.

- Lastly, this research further presents several implications for professionals, building project owners, and contractors on the successes of adopting cloud computing in their projects. This research will also allow all the interested parties to achieve the three critical key success factors of a project in terms of time, cost, and quality by adopting cloud computing on construction activities and finally influencing the overall construction activities' success.

## 9. Limitations and Future Research

Although this research adds to our understanding of the impacts of cloud computing drivers on construction activities, it still has some drawbacks. These concerns include the regional scope of the research. Other limitations are that only construction professionals in Nigeria (Lagos) participated in the survey. Future research should investigate additional areas independently to improve the generalizability of the results. The data analysis was

also based on the responses of 104 respondents; ideally, a larger sample size would have been more suitable. Another statistically significant impact might be observed with a larger sample size. Nevertheless, the small sample size problem was alleviated using the PLS estimation technique. For the sake of this study, the three respondent categories (clients, contractors, and consultants) were all considered to be a single homogeneous group.

Furthermore, since the prediction analysis utilized in this research was restricted to cloud computing drivers, it is recommended that new studies should be organized on the impact of cloud computing obstacles, tools, and applications on construction activities in the future. Finally, the research results on cloud computing implementation requirements and drivers cannot be implemented at the business level in developing companies, and its workers cannot be trained in cloud computing services without the assistance of senior management. Therefore, by refining legislation and standards for adopting cloud computing, the government and top management will promote the implementation of cloud computing in construction projects across the nation.

**Author Contributions:** Research idea, A.F.K.; conceptualization, A.E.O. and A.F.K.; methodology, A.E.O. and A.F.K.; software, A.F.K.; validation, A.F.K., A.E.O., M.M.H., A.S.A. and A.A.; formal analysis, A.F.K.; investigation, A.E.O.; resources, A.E.O.; data curation, A.F.K. and A.E.O.; writing—original draft preparation, A.F.K.; writing—review and editing, A.F.K., A.E.O., M.M.H., A.S.A. and A.A. All authors have read and agreed to the published version of the manuscript.

**Funding:** This research received no external funding.

**Institutional Review Board Statement:** Not applicable.

**Informed Consent Statement:** Not applicable.

**Data Availability Statement:** Not applicable.

**Conflicts of Interest:** The authors declare no conflict of interest.

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
