# Peer review of "Exploring the Cloud Computing Implementation Drivers for Sustainable Construction Projects—A Structural Equation Modeling Approach"

_sustainability, doi:10.3390/su142214789_

Round 1
Reviewer 1 Report
An interesting and well presented topic that it is worth publishing. Really great work. Some only minor comments:
At the introductory section or later the following paper can benefit the research:
Sfakianaki, E. (2019), "Critical success factors for sustainable construction: a literature review", Management of Environmental Quality, Vol. 30 No. 1, pp. 176-196.
In the research model section, it is necessary to address how respondents were selected and justify the sample size. Is it representative of the country’s size?
Define how the different constructs were selected.
Define – I don’t think I saw it – when the research was undertaken.
In terms of English, the paper generally reads well, a good proof reading will help to improve its readability.
Author Response
An interesting and well-presented topic that it is worth publishing. Really great work. Some only minor comments:
Thank you for this very positive comment. We are also grateful for the valuable effort to review the manuscript. Below are the actions taken to embrace your comments:
At the introductory section or later the following paper can benefit the research:
Sfakianaki, E. (2019), "Critical success factors for sustainable construction: a literature review", Management of Environmental Quality, Vol. 30 No. 1, pp. 176-196.
We agree with the reviewer. The above-mentioned valuable reference has been cited. Kindly refer to line 60-61.
In the research model section, it is necessary to address how respondents were selected and justify the sample size. Is it representative of the country’s size?
We are grateful for the constructive comment and fully concur with the reviewer. The justification for the sample size has been added based on the reviewer comment. Kindly refer to lines 337-360:
“The sample size was determined based on the targeted population [112]. Consequently, a suitable statistical analysis technique was chosen to generate the proposed model based on the sample size. SEM was selected for this study; the sample size should be sufficient to achieve the desired result and offer an alternative model [19]. For SEM, Yin [113] agreed that the sample size should be greater than 100.
However, many other researchers oppose maximization and recommend opti-mizing the sample size [114]. They argued that it was not cost-effective and time-efficient at a certain level; however, its generalizability is a significant sample size advantage. Therefore, the minimum measured sample size was taken to achieve the desired statistical power level [115, 116]. The SEM requires a suitable sample size to acquire consistent estimations [117]. Gorsuch [118] suggested a minimum of five participants for every construct and 100 individuals for every data analysis.
Accordingly, the PLS-SEM analysis of the study was the chosen option over the Covariance-Based SEM (CB-SEM) because it best fit the analysis structure of the study. PLS-SEM can be used to evade constricting assumptions that form a total estimate of the total possible deductions with a minimum sample size [119-122]. The sample size for conducting PLS requires only 30-100 responses, as mentioned by [121, 123]. Con-sequently, 137 questionnaires were distributed, and 104 responses were returned and accepted, representing 76 percent of the total population and falling within the ac-ceptable range for analysis [98]. The achieved sample size also meets the minimum numbers recommended, and this tallies the minimum number of sample sizes required [113, 118, 121-123]. Furthermore, the sample size used in this research is similar to that used in a study on applying PLS-SEM in building projects. It suggests that the data gathered was sufficient for future empirical testing [124].”
Define how the different constructs were selected.
Many thanks for the reviewer comment. As mentioned on the paper the constructs renamed based on the previous studies.
Define – I don’t think I saw it – when the research was undertaken.
Many thanks for the reviewer comment. Accordinglly, we set the country context clearly in the introduction section. Kindly refer to lines 93-117:
“This exploratory research outlined the main research question built on the obtained results. The research question is, “What are the relationships and drivers needed to implement cloud computing in the Nigerian construction industry?” Therefore, these relationships need to be examined, and cloud computing drivers too need to be identified [34]. Rockart [35] identifies the drivers as "areas where, if satisfactory, the results will ensure the organization's competitive success." Likewise, Chan, et al. [36] and Yu, et al. [37] agree that the drivers should be seen as important management readiness and action in different construction domains to bring about improvements [38]. By being mindful of these drivers, a firm may favorably impact the success of the development process while successfully mitigating its risks [39].
This research set out to learn what factors in Nigeria's building sector were responsible for its rapid uptake. The current research present a novel attempt to fill this gap using the Partial Least Square (PLS) modeling method to mathematically analyze the relationship between the implementation of cloud computing drivers and construction activities. It is noteworthy that this study used the Global-Local Context (GLC) approach, which emphasized the study's worldwide importance. Besides, it signifies and magnifies the problems examined. Summers [40] adds that establishing the significance of a study in both a local and a broader context is a good method to market its significance. Because of this need for precision, the research will focus on "emerging" nations and, more specifically, on Nigeria as its local context (i.e., establishing the importance). Consequently, this study would be helpful by assisting decision-makers to attain a successful construction project by reducing unnecessary costs and improving efficiency through cloud computing implementation in Nigeria and other underdeveloped nations where similar construction initiatives are being undertaken [41]. Many stakeholders in the construction industry, including policymakers, contractors, and designers, stand to gain from this research [42].”
In terms of English, the paper generally reads well, a good proof reading will help to improve its readability.
We agree with the reviewer. The whole language of the paper has been enhanced according the reviewer’s comments.
Author Response
We are also grateful for the valuable effort to review the manuscript. However, the above mentioned comments are not related to our manuscript. Our paper is related to cloud computing in the construction industry. On the other hand, the above mentioned comments are related to hydroclimate variables under climate change.